# Genetic Basis Identification of a *NLR* Gene, *TaRPM1-2D*, That Confers Powdery Mildew Resistance in Wheat Cultivar ‘Brock’

**DOI:** 10.3390/plants14172652

**Published:** 2025-08-26

**Authors:** Xiaoying Liu, Congying Wang, Yikun Wang, Siqi Wu, Huixuan Dong, Yuntao Shang, Chen Dang, Chaojie Xie, Baoli Fan, Yana Tong, Zhenying Wang

**Affiliations:** 1Tianjin Key Laboratory of Animal and Plant Resistance, College of Life Sciences, Tianjin Normal University, Tianjin 300387, China; skylxy@tjnu.edu.cn (X.L.); 18534997239@163.com (C.W.); wyk_tj@163.com (Y.W.); wusiqi_2024@163.com (S.W.); donghuixuan2024@163.com (H.D.); skyfbl@tjnu.edu.cn (B.F.); 2Tianjin Key Laboratory of Water Resources and Environment, Tianjin Normal University, Tianjin 300387, China; shangyuntao@126.com; 3Key Laboratory of Crop Heterosis and Utilization (MOE), State Key Laboratory for Agro-Biotechnology, Beijing Key Laboratory of Crop Genetic Improvement, China Agricultural University, Beijing 100193, China; dangchensa@163.com (C.D.); xiecj127@126.com (C.X.); 4Tianjin Academy of Agricultural Sciences, Tianjin 300192, China

**Keywords:** *Triticum aestivum* L., *PmBrock locus*, *TaRPM1-2D*, powdery mildew resistance

## Abstract

Wheat powdery mildew, caused by *Blumeria graminis* f. sp. *tritici*, represents one of the most threatening biotic stresses of this crop. The cultivated wheat variety ‘Brock’ exhibits resistance not only to rust but also to powdery mildew, making it a valuable resource for exploitation in wheat disease-resistant breeding. This study identified a novel gene in ‘Brock’ distinct from *Pm2*. In order to identify the disease resistance gene in ‘Brock’, genetic mapping was performed using F_2_ and F_2:3_ populations derived from the cross ‘Jing411/Brock’. The candidate powdery mildew resistance gene was located within a 6.88 Mb physical interval on chromosome 2D, which harbors a highly expressed gene, *TaRPM1-2D*. The protein encoded by *TaRPM1-2D* possesses a typical nucleotide binding, leucine-rich repeat receptor (NLR) domain, and its sequence significantly differs among ‘Jing411’, ‘BJ-1’, and ‘Brock’. Expression of *TaRPM1-2D* was markedly higher in resistant wheat ‘Brock’ and ‘BJ-1’ compared to the susceptible ‘Jing411’. Both overexpression and gene silencing experiments demonstrated that *TaRPM1-2D* contributes to enhance resistance against powdery mildew in wheat. These findings reveal the function of *TaRPM1-2D* in conferring powdery mildew resistance in ‘Brock’ and provide a candidate gene for disease-resistance breeding.

## 1. Introduction

Wheat (*Triticum aestivum* L.) is the cornerstone of global agriculture and a vital staple in human diets, providing 20% of human caloric intake. Wheat production is challenged by biotic stresses, including pathogens and pests that thrive in favorable climates and spread through wind, machinery, and human activities. Among these, wheat powdery mildew, caused by the fungus *Blumeria graminis* f. sp. *tritici* (*Bgt*), is one of the most threatening biological stressors [1]. Under epidemic conditions, yield losses in susceptible varieties can reach 10–15% and up to 50% [2]. The ongoing evolution of the pathogens highlights the long-term challenge of controlling the disease.

Most cloned disease resistance genes can be categorized into two main types of disease resistance proteins: nucleotide binding, leucine-rich repeat receptors (NLRs) and serine-threonine protein kinases [3,4]. NLRs represent a larger and more extensively studied group. The structure of NLR genes is highly diverse, typically featuring a central nucleotide-binding site and a nucleotide-binding adaptor shared by Apaf1, certain R genes and a CED4 (NB-ARC) domain, which are often linked to multiple leucine-rich-repeats (LRRs). As the polymorphism of the aminos end, they may include domains such as the coiled-coil (CNL) subfamily, the Resistance to Powdery Mildew 8 (RNL) subfamily, or the toll/interleukin 1 receptor (TNL) subfamily [5]. Additionally, these proteins may carry various non-canonical integrative domains (IDs), which are directly or indirectly involved in effector protein recognition [6]. To date, 71 powdery mildew disease resistance (*R*) genes have been officially named (*Pm1*–*Pm71*), and many other provisionally named powdery mildew R genes or alleles, of which 19 *R* genes have been cloned [7,8,9,10,11,12,13,14,15,16,17,18,19,20,21,22,23,24,25,26,27]. Among these, 13 *Pm* genes encode CNLs which belong to NLR receptors. In recent years, the *NLR* genes and their synergistic mechanisms have been elucidated, including *Pm3b*/*Pm8*/*Pm17*, *Pm2a*, *Pm21*/*Pm12*, *Pm60*/*MlIW172*/*MlWE18*, *Pm5e*, *Pm41*, *Pm1a*, *Pm69*, *Pm55*, and *Pm6S1*, all of which encode NLR proteins. Research findings have demonstrated that *Pm* genes are distributed across multiple loci on all wheat chromosomes, with chromosome arms 2BS/2BL harboring at least eight *Pm* genes (*Pm6*, *Pm26*, *Pm33*, *Pm42*, *Pm49*, *Pm52*, *Pm63*), establishing it as a hotspot for disease resistance breeding [28]. Additionally, contributions from the Aegilops genus (e.g., *Aegilops geniculata*, *Aegilops tauschii*) have enriched the diversity of resistance sources in cultivated wheat through the introduction of genes such as *Pm12*, *Pm13*, *Pm2*, and *Pm19* [18,22,29]. Furthermore, Jaegle, B et al. employed a k-mer-based genome-wide association studies (GWAS) approach, which not only validated known resistance genes (e.g., *Pm2*, *Pm4*) but also discovered 27 novel potential resistance quantitative trait loci (QTL) [30]. Ramandeep Kaur et al. utilized GWAS to identify 96 candidate genes associated with disease resistance/host–pathogen interactions [31]. However, the co-evolution of wheat with pathogens often leads to the eventual failure of widely deployed resistance genes. To mitigate this problem, the best practice is to integrate multiple *Pm* genes into a single genetic background, which can be achieved by molecular marker-assisted selection using closely linked or gene-specific markers. Therefore, it is necessary to discover, functionally validate, and deploy more and novel resistance genes to expand the gene pool for effective powdery mildew control.

The wheat cultivar ‘Brock’, introduced from the UK, is a valuable genetic resource for rust resistance in wheat breeding. We discovered that it exhibits resistance not only to rust but also to powdery mildew, making it a valuable resource for developing disease-resistant wheat varieties. In our preliminary work, ‘Brock’ served as the donor parent crossed with the recurrent parent ‘Jing411’, which possesses excellent agronomic traits. Through hybridization, backcrossing, and individual selection under powdery mildew stress, the powdery mildew-resistant near-isogenic line ‘Brock/Jing411^7^’, designated as ‘BJ-1’, was developed, achieving the introgression of the powdery mildew resistance gene from ‘Brock’ into ‘BJ-1’ [32]. It is reported that ‘Brock’ carries the powdery mildew resistance gene *Pm2*, located on chromosome 5DS with an NLR structure. *Pm2* and its alleles exhibit significant variation in resistance to different powdery mildew races [9,18,33,34]. Recent studies reported that *Pm2* resistance is being eroded in most major wheat-growing regions of China, probably overcome by pathogen evolution [35]. However, our research on the disease resistance of wheat varieties showed that both ‘Brock’ and ‘BJ-1’ still exhibit significant resistance to powdery mildew. Therefore, we hypothesize that the wheat cultivar ‘Brock’ may contain a novel, previously unidentified resistance gene distinct from *Pm2*, designated *PmBrock*.

To elucidate the structure and function of *PmBrock*, this study conducted gene mapping, cloning, structural and functional analyses of the gene. Virus-induced gene silencing (VIGS) and stable transformation experiments confirmed that *PmBrock* confers powdery mildew resistance in ‘Brock’ and enhances resistance in the near-isogenic line ‘BJ-1’ (developed through ‘Brock’ hybridization).

## 2. Results

### 2.1. Wheat Cultivar ‘Brock’ Contains a New Disease Resistance Gene

PCR amplification using *Pm*-specific primers designed from published sequences revealed both ORF and genomic bands in ‘Jin411’ and ‘Brock’, with notable sequence variations between these cultivars (Appendix A). No amplification products were detected in the near-isogenic line ‘BJ-1’, confirming the absence of *Pm2* in this background. Given that ‘BJ-1’ inherited its powdery mildew resistance from ‘Brock’ and demonstrated effective resistance against *Bgt* in pathogenicity assays, we propose that ‘Brock’ contains at least one additional resistance gene (tentatively designated *PmBrock*) conferring resistance to wheat powdery mildew.

### 2.2. Genetic Analysis of Powdery Mildew Resistance in ‘Brock’

Crossing the susceptible maternal parent ‘Jing411’ with the resistant paternal parent ‘Brock’ generated F_2_ and F_2:3_ genetic populations. Post-inoculation disease assessments at the seedling stage (10 days post inoculation, dpi) revealed complete susceptibility in all ‘Jing411’ plants versus universal resistance in ‘Brock’ (Figure 1). All 36 F_1_ progeny exhibited resistance. Among 250 F_2_ plants screened, 186 demonstrated resistance, while 64 were susceptible, yielding a resistance ratio of 2.91:1. Evaluation of 659 F_2:3_ individuals identified 173 homozygous resistant, 321 segregating, and 165 homozygous susceptible plants, conforming to a 1.05:1.95:1 ratio (Appendix A). These results collectively indicate that powdery mildew resistance in ‘Brock’ is governed by a single dominant gene, which we propose corresponds to the previously postulated *PmBrock*.

### 2.3. Chromosomal Localization of the PmBrock

Initial screening of 1388 public primer pairs using genomic DNA from the maternal parent ‘Jing411’, paternal parent ‘Brock’, and resistant/susceptible bulks of F_2_ and F_2:3_ populations identified 112 polymorphic markers. Subsequent validation with individual F_2_ plants narrowed these to two diagnostic markers: WMC175 and WMC41 (Figure 2). Positional data from the IWGSC database indicated WMC41 resides on chromosome 2D, while WMC175 maps to both 2B and 2D, preliminarily assigning *PmBrock* to chromosome 2D. Given that the known powdery mildew resistance gene Pm43 also localizes to 2D and exhibits the closest genetic linkage to WMC41, the tightly linked SSR marker 2DL9906982 (reported by Liu et al. [36]) was synthesized for fine-mapping. The 2DL9906982 marker displayed consistent polymorphism between resistant and susceptible individuals (Figure 2). According to the PCR marker information on the wheat omics website (http://wheatomics.sdau.edu.cn, accessed on 8 January 2024), six additional polymorphic SSR markers (TC81976, TC85503, Ta.27369.1, CA615959, CA695916, and GPW2172) were developed between 2DL9906982 and WMC175. Seedling disease resistance phenotypes of resistant/susceptible plants were integrated with the SSR amplification results, and genetic linkage analysis was performed using Joinmap 4.0. *PmBrock* was flanked by CA695916 and GPW2172 (Figure 2), corresponding to genetic distances of 1.7 cM and 3.1 cM, respectively (Figure 3). *PmBrock* was further mapped to a 6.88 Mb physical interval (chr2D: 596,254,451–603,132,553 bp).

### 2.4. Genomic Information Within the Disease Resistance Interval

Based on the delimited physical interval, alignment with the Chinese Spring genome assembly in the IWGSC RefSeq database annotated 212 candidate genes. Comprehensive genomic features—including physical positions, gene identifiers, and functional predictions—were systematically cataloged (Appendix A). The related genes participate in diverse biological processes: biotic and abiotic stresses, post-transcriptional regulation, translational control, plant retrotransposon activity, DNA replication, developmental regulation, transmembrane transport, and photosynthetic mechanisms. Overall 29 genes encode kinase-related proteins, primarily involved in signal transduction pathways. Among these candidate genes, 11 encode disease resistance proteins, of which six contain NBS-LRR domains (*TraesCS2D01G503500*, *TraesCS2D01G504500*, *TraesCS2D01G626800LC*, *TraesCS2D01G510000*, *TraesCS2D01G510100*, *TraesCS2D01G510300*). Previous RNA-seq analysis of the powdery mildew-resistant wheat ‘Brock’ infected with *Bgt E09* identified *TraesCS2D01G504500* as significantly up-regulated post-infection [37]. Therefore, we selected this gene for further analysis.

### 2.5. Cloning and Structural Characterization of TaRPM1-2D

The full-length sequence of *TraesCS2D01G504500* (designated *TaRPM1-2D*) was amplified based on the Chinese Spring genome assembly. The open reading frame (ORF) of *TaRPM1-2D* comprises 2754 base pairs (bp), predicted to encode a protein of 917 amino acids with a predicted molecular weight of 105.48 kDa (Appendix A). Phylogenetic analysis revealed high homology with other RPM1 from several wheat cultivars, indicating conserved functional domain (Figure 4).

### 2.6. Expression Dynamics of TaRPM1-2D Under Powdery Mildew Stress

In resistant cultivar ‘Brock’ and NILs, *TaRPM1-2D* exhibited rapid transcriptional induction at early infection stages (2–4 hpi). In contrast, the susceptible cultivar ‘Jing411’ showed delayed response until 12 hpi, with substantially lower expression magnitude. Expression surged at 4 hpi, transiently declined at 8 hpi (near baseline), then progressively increased to peak levels at 48 hpi in ‘Brock’. Post-peak expression remained elevated above controls throughout 72 hpi–7 dpi. In NILs, base expression (0 hpi) was intermediate between ‘Brock’ and ‘Jing 411’. Upregulation initiated at 2 hpi, peaked at 12 hpi (exceeding ‘Brock’’s concurrent levels), and gradually returned to baseline by 7 dpi. In ‘Jing411’, minimal base expression was detected. Despite a modest peak at 12 hpi, transcript levels consistently trailed resistant lines across all timepoints (0–7 dpi), with subsequent fluctuations remaining below resistant genotypes (Figure 5).

### 2.7. Functional Validation of TaRPM1-2D Using VIGS

Three control groups (BSMV:PDS, BSMV:GFP, and GKP-Buffer solution) and one experimental group BSMV:*TaRPM1* were established. Recombinant virus were mixed per standardized silencing protocols and rub-inoculated onto the second fully expanded leaves of ‘Brock’ wheat. RT-qPCR analysis confirmed significantly reduced *TaRPM1-2D* transcript levels in BSMV:*TaRPM1* plants compared to both controls throughout the experiment (Appendix A). Post-infection with *Bgt* at 7 dpi, BSMV:*TaRPM1* leaves exhibited extensive powdery mildew colonies, whereas GKP-Buffer and BSMV controls showed only sparse, isolated microcolonies (Figure 6A). Coomassie brilliant blue staining revealed prolific hyphal networks and conidiophores in BSMV:*TaRPM1* tissues, contrasting with abnormal appressorium and limited hyphae in controls (Figure 6B). BSMV:*TaRPM1* plants showed 5.16-fold and 2.33-fold higher fungal penetration rates than GKP-Buffer and BSMV:*GFP*, respectively (Figure 6C, Appendix A); abnormal appressoria peaked in BSMV:GFP (17.2%), followed by GKP-Buffer (12.8%), with BSMV:*TaRPM1* having the lowest deformity rate (8.3%), (Figure 6D, Appendix A); hypersensitive reaction cells were significantly reduced in BSMV:*TaRPM1* (5.8‰) versus controls (GKP-Buffer: 11.12‰; BSMV:*GFP*: 8.46‰, Figure 6E, Appendix A). Collectively, silencing *TaRPM1-2D* severely compromised ‘Brock’’s resistance to powdery mildew, validating its essential role in disease defense.

### 2.8. Functional Validation of TaRPM1-2D via Overexpression Analysis

Primers designed according to the ubi promoter and *TaRPM1-2D* sequence amplified an expected ~750 bp fragment from transgenic wheat lines. Genomic DNA PCR screening included negative controls (wild-type ‘Jinqiang 5’, lane 3) alongside a positive control (pTCK305-TaRPM1 plasmid, lane 2) (Appendix A). Three putative transgenic lines were verified as positive. Quantitative RT-PCR with actin as the reference gene demonstrated elevated *TaRPM1-2D* expression in transgenic ‘Jinqiang 5’ compared to wild-type plants (Appendix A). OE-1 exhibited the strongest expression (~5-fold increase), while other lines showed 2–3 fold upregulation.

T_1_ transgenic lines (OE-1, OE-3, and OE-4) were challenged with *Bgt* at the one-leaf stage. At 5 dpi, wild-type ‘Jinqiang 5’ developed extensive powdery mildew colonies, whereas the transgenic line, particularly OE-1, displayed significantly smaller and sparser colonies (Figure 7A,C). This reduction of powdery mildew colony density confirmed that *TaRPM1-2D* overexpression enhances powdery mildew resistance in the otherwise susceptible cultivar (Figure 7B, Appendix A), establishing its role as a positive regulator of disease defense.

## 3. Discussion

Wheat powdery mildew, caused by *Bgt*, is a devastating disease severely threatening global wheat production. For years, yield losses attributed to this disease have reached 10–15%, and can escalate to 50% under severe epidemics [2]. Due to the strong selection for virulent pathogen mutants under agricultural conditions, the emergence and spread of new highly virulent pathogen strains have led to the gradual breakdown of resistance in bred and deployed wheat varieties [38]. Consequently, the continuous development and deployment of powdery mildew-resistant cultivars remain critical objectives in wheat genetic improvement. To broaden and enrich resistance sources, the ongoing identification and cloning of novel disease resistance genes from commercial wheat varieties, landraces, wild relatives, and distant species provide essential genetic resources for resistance breeding.

The wheat cultivar ‘Brock’, introduced from the UK as a genetic resource for rust resistance, exhibits additional resistance to powdery mildew, making it a valuable resource for disease-resistant breeding. In preliminary work, the near-isogenic line (NIL) ‘BJ-1’ was developed using ‘Brock’ (donor) and the elite cultivar ‘Jing411’ (recurrent parent) [32]. ‘BJ-1’ not only surpasses ‘Jing411’ in powdery mildew resistance but also exhibits superior agronomic performance. Specifically, key traits including grain yield and tiller number demonstrate significant enhancement in ‘BJ-1’ [39]. Its improved disease resistance profile contributes to diminished pathogen-related damage, thereby substantially lowering the probability of disease-induced yield reductions. Consequently, investigating the disease resistance gene of ‘BJ-1’ (‘Brock’) against *Blumeria graminis* f. sp. *tritici* (*Bgt*) is scientifically imperative. Although earlier studies reported the presence of the *Pm2* gene in ‘Brock’ [9,33,34], recent research indicated that *Pm2* and its alleles have lost effectiveness against prevalent powdery mildew. Notably, both ‘Brock’ and ‘BJ-1’ retain strong resistance to *Bgt*. PCR amplification of the *Pm2* gene in ‘Brock’, ‘Jing411’, and ‘BJ-1’ revealed the full-length *Pm2a* sequence exclusively in ‘Brock’. The absence of this sequence in ‘Jing411’ and ‘BJ-1’, despite ‘Brock’ being the donor parent for ‘BJ-1’, suggests that ‘Brock’ harbors at least one resistance gene distinct from *Pm2*.

To dissect the resistance gene(s) in ‘Brock’, genetic mapping was performed using F_2_ and F_2:3_ populations derived from the cross “Jing 411 × Brock”. The candidate gene was mapped to a 4.8 cM genetic interval flanked by markers WMC41 and GPW2172, corresponding to a 6.88 Mb physical region on chromosome 2D in the Chinese Spring genome. This interval contains six *NBS-LRR* genes. According to previous reports, there was a *Pm43 locus* on chr 2D [36]. Comparative analysis found that the *PmBrock* region overlaps with the *Pm43 locus*. Integration of mapping data confirmed that both the *PmBrock* and *Pm43* intervals comprise multiple *NBS-LRR* genes, with a shared region harboring five *NBS-LRR* genes. This indicates a potential *NLR* gene cluster where multiple resistance genes collectively confer powdery mildew resistance. Research on disease gene function in the *PmBrock locus* may clarify collaborative mechanisms for varietal enhancement, supporting sustainable wheat breeding development.

Online expression analysis indicated that *TraesCS2D01G504500* (*TaRPM1-2D*) displays higher expression in wheat leaves compared to other genes in this region. Previous RNA-seq analysis by our group also detected significant upregulation of *TaRPM1-2D* in ‘Brock’ during early *Bgt* infection, supporting its role as a candidate resistance gene. Cloning of *TaRPM1-2D* from ‘Brock’, ‘BJ-1’, and ‘Jing411’ revealed distinct sequence differences among two varieties, while maintaining a canonical NLR domain structure. Expression levels of *TaRPM1-2D* were significantly higher in resistant lines (‘Brock’ and ‘BJ-1’) than in the susceptible ‘Jing411Jing’. Both overexpression and gene silencing experiments confirmed that *TaRPM1-2D* expression enhances resistance to *Bgt*. Phenotypic consistency among independent transgenic events and VIGS constructs verifies that *TaRPM1-2D* acts as a positive regulator of wheat immunity toward *Bgt*, which had not been previously characterized in wheat’s resistance to powdery mildew. These results demonstrate that *TaRPM1-2D* confers the powdery mildew resistance of the *PmBrock locus*.

Extensive breeding studies consistently reveal an inverse correlation between pathogen resistance and agricultural productivity. Plant varieties exhibiting robust disease resistance often demonstrate compromised growth vigor and reduced yield potential. Conversely, high-yielding cultivars tend to display diminished resistance to pathogenic threats [40]. Preliminary field evaluations of the NIL ‘BJ-1’ (carrying *TaRPM1-2D*) showed no significant negative impact on agronomic or yield traits under disease-free conditions, highlighting its potential value in breeding. Gene pyramiding is an effective strategy for achieving durable and broad-spectrum resistance against plant diseases [41,42]. Therefore, the strategic integration of *TaRPM1-2D* into high-quality wheat cultivars holds promise for enhancing powdery mildew resistance and contributing to sustainable global wheat production.

## 4. Materials and Methods

### 4.1. Plant Materials

‘Jing411’ is a wheat variety with excellent agronomic traits but susceptible to powdery mildew. ‘Brock’ is a wheat variety resistant to powdery mildew, kindly provided by Professor Yang Zuomin from China Agricultural University. For ‘BJ-1’, in preliminary work, disease-resistant wheat ‘Brock’ was used as the donor parent crossed with susceptible wheat ‘Jing411’ to obtain F_1_ progeny. With susceptible wheat ‘Jing411’ as the recurrent parent, after six consecutive backcross generations followed by one selfing generation, disease-resistant individuals were selected under powdery mildew infection pressure at each generation, resulting in the powdery mildew-resistant near-isogenic wheat line ‘BJ-1’. Disease resistance evaluation confirmed that ‘BJ-1’ exhibits resistance to powdery mildew. For the “Jing411/Brock” genetic population, with wheat ‘Jing411’ as the female parent and ‘Brock’ as the male parent, F_1_ seeds were obtained through hybridization. F_1_ plants were sown in the field to harvest F_2_ seeds. F_2_ plants were then individually sown to harvest F_2:3_ seeds. ‘Jinqiang5’ is a spring wheat variety with excellent agronomic traits but susceptible to powdery mildew, which was taken as receptor wheat during overexpression.

### 4.2. Evaluation of Disease Resistance in Wheat Populations

Seeds of 1005 individuals from the “Jing411/Brock” genetic population individuals, ‘Jing411’, and ‘Brock’ were uniformly sown at a density of five seeds per well in 72-well trays. The trays were incubated under controlled conditions: 16 h light, 8 h dark, and a constant temperature of 25 °C. Plants were grown until the one-leaf stage, then exposed to a powdery mildew pathogen through inoculation via the dusting method and incubated for 10 days. Seedling resistance to powdery mildew was subsequently evaluated using image-based software (Image J software v 1.8.0). After about 10 dpi, the infection type of the materials was investigated according to the grading standard of 0~4. The experiments were conducted simultaneously, and repeated three times.

### 4.3. Sequence Alignment Analysis of Pm2 Gene in ‘Jing411’, ‘Brock’, and Near-Isogenic Line ‘BJ-1’

Primers targeting the *Pm2* gene were designed based on genomic data from Chinese Spring wheat (all primer sequences detailed in Appendix A). PCR amplification was performed on genomic DNA of ‘Jing411’, ‘Brock’, and the near-isogenic line ‘BJ-1’ using these primers. PCR conditions were conducted in a 25 uL volume: 5× PSGXL Buffer 5.00 uL, dNTP Mix 2.00 uL, cDNA 1.00 uL, PrimeSTAR GXL DNA Polymerase 0.50 uL, Pm2-F1 (Pm2-F2) 0.50 uL, and Pm2-R1 (Pm2-R2) 0.50 uL. Amplified Pm2 sequences were aligned using DNAMAN software v 9.0 to analyze sequence variations among the three materials.

### 4.4. Construction of the PmBrock Genetic Linkage Map

Initially, ‘Jing411’ and ‘Brock’ were screened. We selected 1388 pairs of simple sequence repeats (SSR) polymorphic molecular markers covering the whole wheat genome. The selected 848 primers were then used to further screen the F_2_ resistant and susceptible pools, followed by a more detailed screening of the F_2:3_ resistant and susceptible pools. Polymorphic markers identified from bulk screening were used for individual plant genotyping. After identifying the differential primers, the specific interval was delimited, and the SSR markers within this interval were identified and synthesized to further narrow down the region of the disease resistance gene. JoinMap4.0 software was used to calculate genetic distances, and Mapchart was employed to visualize the genetic linkage map [43].

### 4.5. Analysis of Candidate Genes Within the Disease Resistance Interval

Using the genetic linkage map and the disease data of the genotypes, the disease resistance gene interval was located. Candidate genes were sought by referencing the Chinese Spring genome information in the IWGSC database of WheatOmics 1.1 (http://202.194.139.32/, accessed on 8 January 2024). Relevant gene information was downloaded, including gene expression profile data, which provided insights into the expression patterns of these candidate genes in wheat roots, stems, leaves, spikes, and grains. Subsequently, based on gene descriptions and existing research, the functions of these genes were initially predicted, and the genes were classified accordingly.

### 4.6. Cloning of TaRPM1-2D

Primers were designed based on the *TraesCS2D01G5034500* sequence to amplify the target gene through upstream and downstream sequences (primer sequences detailed in Appendix A). The conserved domains of TaRPM1-2D were analyzed in NCBI/Blastp, and the high homology proteins were downloaded and then used to construct phylogenetic trees using DNAMAN software.

### 4.7. Analysis of TaRPM1-2D Expression Patterns

Primer pair qPCR-RPM1-N-F/R was designed based on sequences within the non-conserved region of the gene. Total RNA was extracted from frozen-stored wheat leaves infected with *Bgt* at ten different time points (Figure 5) and reverse-transcribed into cDNA for use as a template. Actin served as the internal reference. RT-qPCR was employed to validate *TaRPM1-2D* gene expression patterns across different wheat cultivars (*Triticum aestivum* L.). The experiment was conducted following the manufacturer’s protocol for FastStart Universal SYBR Green Master (ROX) (Roche, Basel, Switzerland).

### 4.8. Functional Analysis of TaRPM1-2D Gene Using VIGS

Based on the *TaRPM1-2D* gene sequencing results, a primer pair was designed targeting the non-conserved region for gene silencing. *Nhe* I restriction sites were incorporated flanking the target gene, which was then ligated into the *Nhe* I-digested and purified BSMVγ vector backbone to construct the recombinant plasmid BSMVγ:*TaRPM1*. The recombinant vectors BSMVα, BSMVβ, BSMVγ:*PDS*, BSMVγ:*GFP*, and BSMVγ:*TaRPM1* were linearized and transcribed into RNA in vitro. Mixtures containing BSMVα, BSMVβ, and either BSMVγ:*PDS*, BSMVγ:*GFP* or BSMVγ:*TaRPM1* (1:1:1 ratio) were rub-inoculated onto ‘Brock’ leaves. Leaves inoculated with BSMVγ served as the experimental group, while those receiving BSMVγ:*GFP* or GKP buffer constituted the control groups. After emergence of the third leaf, plants were inoculated with *Bgt*. Powdery mildew spore development was observed at 7 dpi, with infection efficiency and malformation rate quantified. Phenotypic changes in VIGS-treated wheat leaves were systematically recorded.

### 4.9. Functional Analysis of TaRPM1-2D Gene via Overexpression

The *TaRPM1-2D* gene was amplified with pTCK303 vector adapters and ligated into the *Kpn* I/*Spe* I-digested and purified pTCK303 backbone to construct the overexpression recombinant vector. This recombinant plasmid was transformed into *Agrobacterium tumefaciens* strain EHA105, and *TaRPM1-2D* was subsequently introduced into wheat cultivar ‘Jinjiang 5’ using the *Agrobacterium*-mediated mature embryo transformation system [44]. Transgenic wheat lines were confirmed through genomic PCR and qRT-PCR screening. At the one-leaf stage, transgenic plants were infected with *Bgt* for 7 dpi, and the growth of powdery mildew on the surface of transgenic wheat leaves was observed.

## 5. Conclusions

In this study, a new powdery mildew *locus*, *PmBrock locus*, was located on chromosome 2DL. Six *NLRs* genes, including *TaRPM1-2D,* had higher expression in ‘Brock’. OE and VIGS results indicated that *TaRPM1-2D* confers *Bgt* resistance in ‘Brock’ and enhances resistance in the near-isogenic line ‘BJ-1’. This study establishes that *TaRPM1-2D* expands current understanding of NLR protein multifunctionality, while offering concrete genetic targets for developing wheat cultivars with enhanced *Bgt* resistance.

## Figures and Tables

**Figure 1 plants-14-02652-f001:**
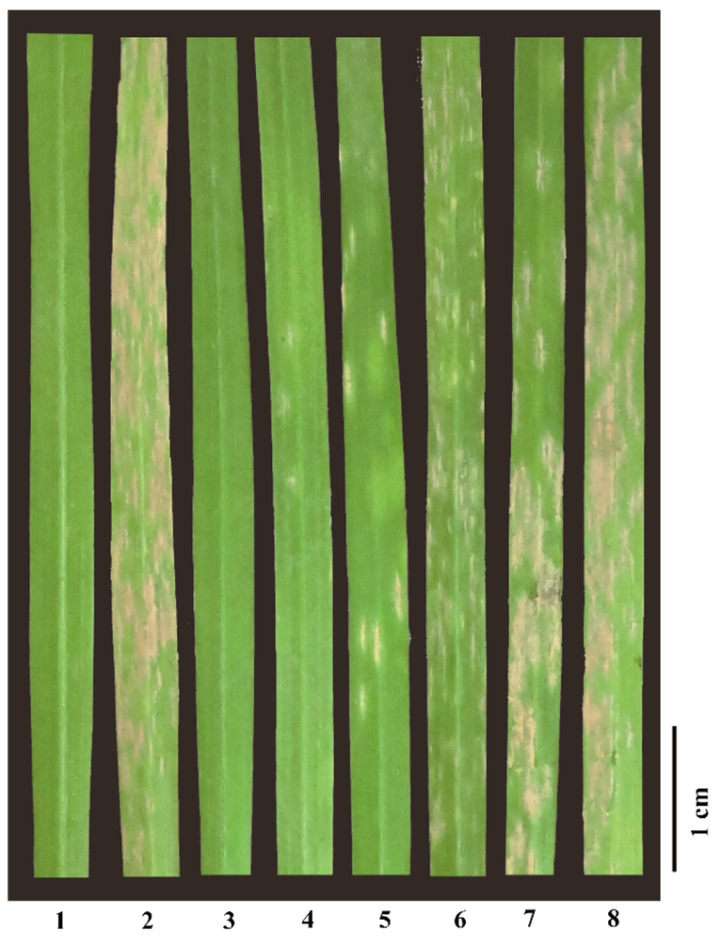
The seedling response of F_2_ populations and its parents to *Blumeria graminis* f. sp. *tritici* (*Bgt*) *E09*. 1: ‘Brock’ (IT = 0), 2: ‘Jing411’ (IT = 4), 3: F_2_ individuals (IT = 0), 4: F_2_ individuals (IT = 0′), 5: F_2_ individuals (IT = 1), 6: F_2_ individuals (IT = 2), 7: F_2_ individuals (IT = 3), 8: F_2_ individuals (IT = 4). Seedling IT values of 0–2 indicate resistance and 3–4 susceptibility.

**Figure 2 plants-14-02652-f002:**
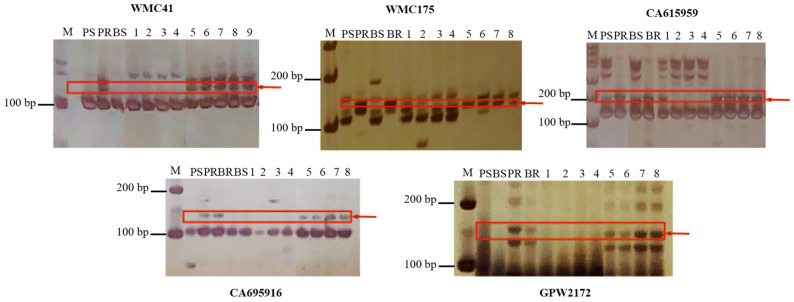
PCR-amplification with SSR markers in parents, resistant pool, and F_2_ individuals. M: DL 1000 Marker, PS: susceptible parent ‘Jing411’, PR: resistant parent, BS: susceptible pool, BR: disease-resistant pool, 1–4: susceptible individual plants; 5–9: disease-resistant individual plants. Red arrow pointed to the specific PCR-amplification with SSR marker in disease resistant samples.

**Figure 3 plants-14-02652-f003:**
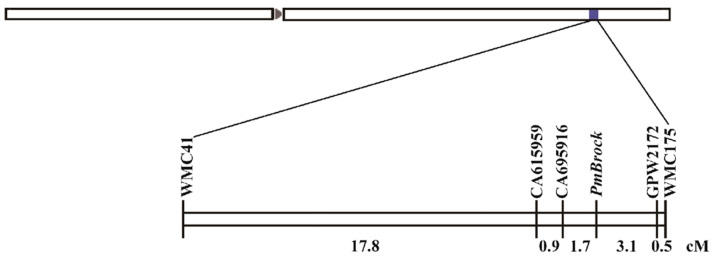
Genetic linkage map and physical location map of *PmBrock locus*. *PmBrock* flanked by CA695916 and GPW2172, corresponding to genetic distances of 1.7 cM and 3.1 cM, respectively.

**Figure 4 plants-14-02652-f004:**
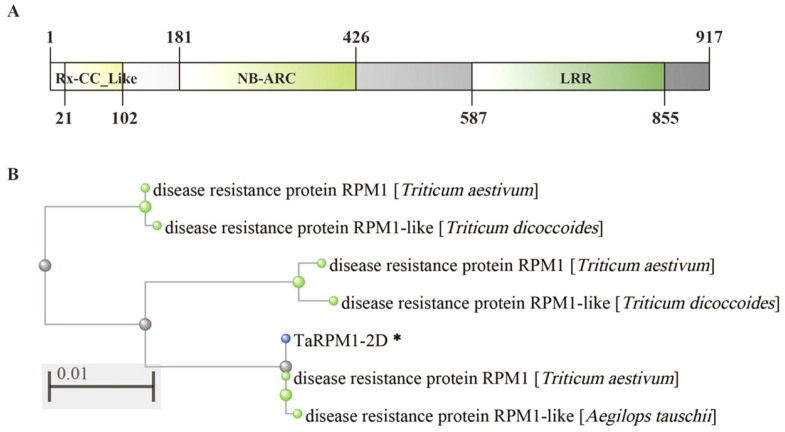
Structure characteristic and phylogenetic analysis of TaRPM1-2D. (**A**) Predicted structure of TaRPM1-2D. (**B**) Phylogenetic tree of TaRPM1-2D in several hexaploid wheats and durum wheat. Light yellow: RX-CC_Like domain; light green: nucleotide-binding adaptor shared by Apaf1, certain R genes and CED4 (NB-ARC) domain; green: leucine-rich-repeat (LRR) domain. * represents the candidate protein.

**Figure 5 plants-14-02652-f005:**
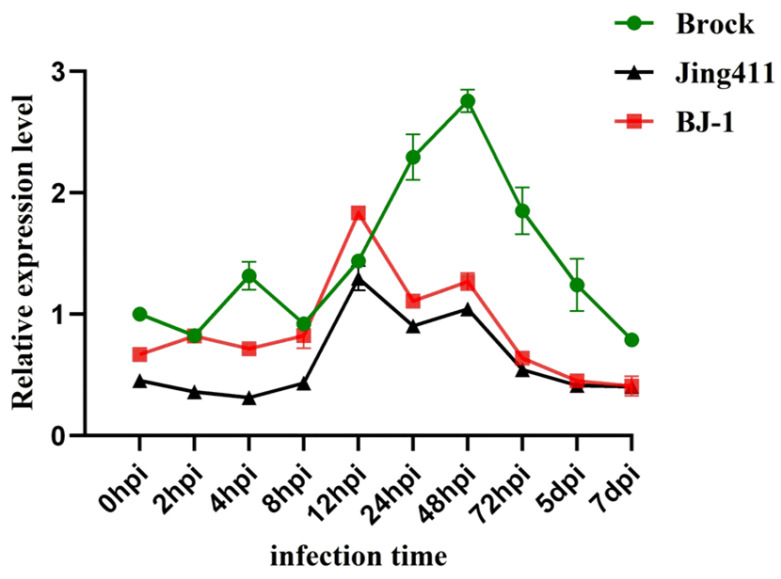
Expression analysis of *TaRPM1-2D* after *Bgt* infection.

**Figure 6 plants-14-02652-f006:**
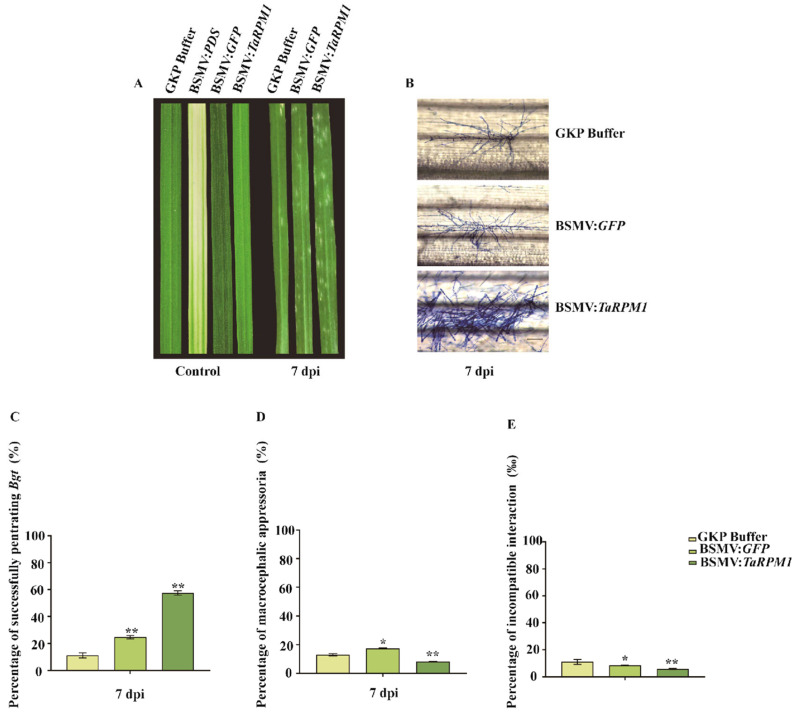
Silencing of *TaRPM1-2D* reduces ‘Brock’ resistance against *Bgt E09* at 7 dpi. (**A**) Phenotype of silenced leaves. (**B**) *Bgt* conidia on *TaRPM1-2D* silenced leaves. (**C**) Percentage of successfully penetrating by *Bgt*. (**D**) Percentage of macrocephalic appressoria. (**E**) Percentage of hypersensitive reactions in *TaRPM1-2D* silenced leaves. (* *p* < 0.05, ** *p* < 0.01).

**Figure 7 plants-14-02652-f007:**
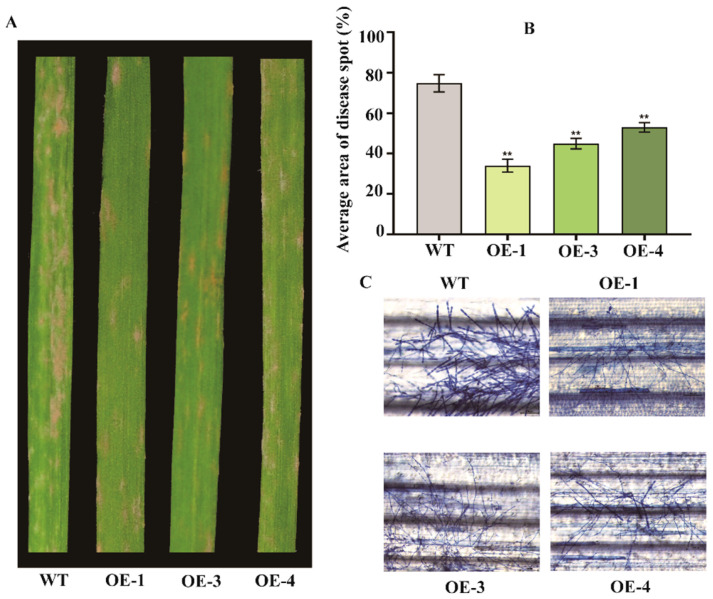
Over-expression of *TaRPM1-2D* in susceptible wheat variety ‘Jinqiang5’. (**A**) Phenotype of *TaRPM1-2D* OE and control wheat leaves inoculated with *Bgt E09* for 7 dpi. (**B**) Lesion area of OE and control wheat leaves. (**C**) *Bgt* conidia on *TaRPM1-2D* OE leaves. WT: ‘Jinqiang5’; OE-1,3, 4: transgenic ‘Jinqiang5’ wheat containing the recombinant vector pTCK303:*TaRPM1-2D* (** *p* < 0.01).

## Data Availability

Data are contained within the article or Appendix A.

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
