# Peer review of "Genetic Basis Identification of a NLR Gene, TaRPM1-2D, That Confers Powdery Mildew Resistance in Wheat Cultivar ‘Brock’"

_plants, 2025, doi:10.3390/plants14172652_

Round 1

Reviewer 1 Report

Comments and Suggestions for Authors

Genetic Basis Identification of a NLR Gene, TaRPM1-2D, That Confers Powdery Mildew Resistance in Cultivar Wheat Brock

2025/8/12

 This study identified a novel gene in Brock distinct from Pm2. In order to identify the disease resistance gene in Brock, genetic mapping was performed using F2 and F2:3 populations derived from the cross "Jing411/Brock". These findings reveal the function of TaRPM1-2D in conferring powdery mildew resistance in Brock and provide a candidate gene for disease-resistant breeding. Following considerable revision, the journal may consider the manuscript. Please find below some comments that will improve the paper.

Comments and questions to improve:

(1)The layout of figure(Figure 2) and tables seems a little bit messy to me. The text font size in some figures (Figure 6) is too small to be read clearly.
(2)The author should discuss and include the reference and point out the differences. The problem should be stated more clearly and in more detail, with more supporting references. 
(3)The author should provide some comparison graphs of the results. Please include a comparative analysis.
(4)What are the development prospects of the study? This is important because PmBrock confers powdery mildew resistance in Brock and enhances resistance in the near-isogenic line BJ-1 (developed through Brock hybridization), the authors point out.
(5)The conclusion is very short, and the results should be better discussed. Finally, the directions of further research should be mentioned in the conclusion section.

Author Response

Dear Reviewer,

Thanks very much for taking your time to review this manuscript. I really appreciate all your comments and suggestions! Please find my itemized responses in below and my revisions/corrections in the re-submitted files.

Sincerely,

Zhenying Wang

Appended to this letter is our point-by-point response to the comments raised by the reviewers. All changes in the text are highlighted in different colors.

(1)The layout of figure(Figure 2) and tables seems a little bit messy to me. The text font size in some figures (Figure 6) is too small to be read clearly.

Answer: Thank you for the reviewer’s suggestion, we have modified the figure 2 through adding red frames. And we revised the font size in figure 6 to optimize the image. The figures revised were all re-submitted. The changes were highlighted in green color.

(2)The author should discuss and include the reference and point out the differences. The problem should be stated more clearly and in more detail, with more supporting references.

Answer: Thank you for the reviewer’s suggestion, we have modified the description in introduction and discussion part, and added new references. The changes were highlighted in green, purple, and orange color.

(3)The author should provide some comparison graphs of the results. Please include a comparative analysis.

Answer: According to the reviewer’s suggestion, we added a supplementary table S4 to illustrate the TaRPM1-2D function. More comparative analysis were stated in diucussion part Lin2329-331. The changes were highlighted in green color.

(4)What are the development prospects of the study? This is important because PmBrockconfers powdery mildew resistance in Brock and enhances resistance in the near-isogenic line BJ-1 (developed through Brock hybridization), the authors point out.

Answer: Thank you for the reviewer’s suggestion, we have added the comments in paragraph 2 and 5 of discussion. The detailed contents: “'BJ-1' not only surpasses 'Jing411' in powdery mildew resistance but also exhibits superior agronomic performance. Specifically, key traits including grain yield and tiller number demonstrate significant enhancement in 'BJ-1'. Its improved disease resistance profile contributes to diminished pathogen-related damage, thereby substantially lowering the probability of disease-induced yield reductions. Consequently, investigating the disease resistance gene of ‘BJ-1’ ('Brock') against Blumeria graminis f. sp. tritici (Bgt) is scientifically imperative.” in paragraph 2; “Extensive breeding studies consistently reveal an inverse correlation between pathogen resistance and agricultural productivity. Plant varieties exhibiting robust disease resistance often demonstrate compromised growth vigor and reduced yield potential. Conversely, high-yielding cultivars tend to display diminished resistance to pathogenic threats.” in paragraph 5. The changes were highlighted in green color.

(5)The conclusion is very short, and the results should be better discussed. Finally, the directionsof further research should be mentioned in the conclusion section.

Answer: According to the reviewer’s suggestion, we revised the description in conclusion. The changes were highlighted in green color.

Reviewer 2 Report

Comments and Suggestions for Authors

In this study the Authors identified a new gene conferring poedery  mildew-resistance in the wheat cultivar Brock, using genetic mapping of a F2 and F3 progenies of the cultivar Block and the powdery -mildew susceptible cultivar Jing 414.

The state of the art and the research objective are clearly indicated, the experimental design is appropriate and the results are clearly illustrated and commented.

I have only minor concerns, mostly formal:

  • Abstract Line 27: spell the terms in an extended form when using an acronym for the first time in the text.
  • -Introduction Line 41 : do not use a  caèital letter for the name of a species or forma specialis.
  • Introduction Lines 49 and 50: spell the terms in an extended form when using an acronym for the first time in the text.
  • Results Line 91-92: this sentence has to be deleted from the Results.
  • Results Lines 159 and 219: do not initiate a sentence with a number.
  • Results Lines 169-171: there is something wrong with this sentence; please, rephrase.
  • Discussion 262: there is something missing in this sentence; please, check.
  • Troughouth the text, cite the name of cultivars between quotation marks if it is not accompained by the term cultivar (e.g. 'Brock', 'Jing 411'  etc. or cultivar Brock, cultivar Jing 411 etc.

For other minore text editings or typos see notes in the text (attached PDF file).

Author Response

Dear Reviewer,

Thanks very much for taking your time to review this manuscript. I really appreciate all your comments and suggestions! Please find my itemized responses in below and my revisions/corrections in the re-submitted files.

Sincerely,

Zhenying Wang

Appended to this letter is our point-by-point response to the comments raised by the reviewers. All changes in the text are highlighted in different colors.

(1) Abstract Line 27: spell the terms in an extended form when using an acronym for the first time in the text.

Answer: Thank you for the reviewer’s suggestion, we have modified the terms. The detailed contents were “ nucleotide binding, leucine-rich repeat receptor (NLR) ”. The other contents were all revised. The changes were highlighted in orange color.

(2) Introduction Line 41 : do not use a caèital letter for the name of a species or forma specialis.

Answer: Thank you for the reviewer’s suggestion, we have modified the letter. The changes were highlighted in orange color.

(3) Introduction Lines 49 and 50: spell the terms in an extended form when using an acronym for the first time in the text.

Answer: According to the reviewer’s suggestion, we have modified the terms. The detailed contents were “nucleotide-binding adaptor shared by Apaf1, certain R genes and CED4 (NB-ARC) domain”, and “leucine-rich-repeats (LRRs)”. The changes were highlighted in orange color.

(4) Results Line 91-92: this sentence has to be deleted from the Results

Answer: According to the reviewer’s suggestion, we have deleted that sentence.

(5) Results Lines 159 and 219: do not initiate a sentence with a number Results

Answer: According to the reviewer’s suggestion, we have modified the description. The changes were highlighted in orange color.

(6) Lines 169-171: there is something wrong with this sentence, please, rephrase

Answer: We have rephrased the sentence, the detailed contents were “The open reading frame (ORF) of ‌TaRPM1-2D‌ comprises ‌2754 base pairs (bp)‌, predicted to encode a protein of ‌917 amino acids‌ with a predicted molecular weight of ‌105.476 kDa (Figure S2).” The changes were highlighted in orange color.

(7) Discussion 262: there is something missing in this sentence, please, check Through the text, cite the name of cultivars between quotation marks if it is not accompainedby the term cultivar (e.g. 'Brock', 'Jing 411' etc. or cultivar Brock, cultivar Jing 411 etc.For other minore text editings or typos see notes in the text (attached PDF file)

Answer: We have revised the sentence, the detailed contents were “According to Liu et al. reported, there was a Pm43 locus on chr 2D. Comparative analysis found that PmBrock region overlaps with the Pm43 locus.” 

    We have cited the name the name of cultivars between quotation, which weren’t  accompainedby the term cultivar. The other minore text editings and typos were all revised in the text according to the reviewer’s suggestions.

    The changes were highlighted in orange color.

Reviewer 3 Report

Comments and Suggestions for Authors

This paper provides excellent expression analysis of the TaRPM1-2D gene in the cultivated wheat variety "Brock," which is involved in powdery mildew resistance.

I think it would be worth considering the following points.

  1. There is no explanation of how the genetic linkage map and physical location map in Figure 3 were created. There is also no explanation of this figure in the main text, so I think it would be better to include an explanation in the legend and describe the interpretation of the results.
  2. I don't know what software was used for the phylogenetic analysis in Figure 4-B. I think it would be better to clearly describe what was used for the phylogenetic analysis. I also think it would be better to publish the protein amino acid sequences used to draw this figure as a supplementary file.
  3. I think the analysis of expression experiments is good.

I don't think this paper has major problems and grammatical problems.

Author Response

Dear Reviewer,

Thanks very much for taking your time to review this manuscript. I really appreciate all your comments and suggestions! Please find my itemized responses in below and my revisions/corrections in the re-submitted files.

Sincerely,

Zhenying Wang

Appended to this letter is our point-by-point response to the comments raised by the reviewers. All changes in the text are highlighted in different colors.

(1) There is no explanation of how the genetic linkage map and physical location map in Figure 3 were created. There is also no explanation of this figure in the main text, so I think it would be better to include an explanation in the legend and describe the interpretation of the results.

Answer: Thank you for the reviewer’s suggestion, we have modified the description about linkage map and physical location map. The detailed contents were “Integrating seedling disease resistance phenotypes of resistant/susceptible plants with SSR amplification results, genetic linkage analysis was performed using Joinmap 4.0. PmBrock flanked by CA695916 and GPW2172 (Figure 2), corresponding to genetic distances of 1.7 cM and 3.1 cM, respectively (Figure 3). And PmBrock was further mapped to a 6.88 Mb physical interval (chr2D: 596254451–603,132,553 bp) .” in results and legend. The changes were highlighted in red color.

(2) I don't know what software was used for the phylogenetic analysis in Figure 4-B. I think it would be better to clearly describe what was used for the phylogenetic analysis. I also think it would be better to publish the protein amino acid sequences used to draw this figure as supplementary file.

Answer: Thank you for the reviewer’s suggestion, we have revised the description in 4.6. The detailed contents were “The conserved domains of TaRPM1-2D were analyzed in NCBI/Blastp, and downloaded the high homology proteins, which were used to construct phylogenetic trees using DNAMAN software.” We also add amino acid sequences figure as supplementary file.The changes were highlighted in red color.

(3) I think the analysis of expression experiments is good

don't think this paper has major problems and grammatical problems.

Answer: We sincerely appreciate the reviewers' positive feedback.

Reviewer 4 Report

Comments and Suggestions for Authors

Dear Authors,

The MS about the novel powdery mildew gene on chromosome 2D makes a nice story and could be a new addition to scientific knowledge in the field of plant breeding as well as plant pathology.

I have made several comments in the MS (the file is being attached for your kind perusal). Those need to be addressed before could be considered from my side for acceptance.

On the other hand, I have some specific comments that might or might not be part of your MS but needs some addressing keeping in view the context of the article. 

Line 71: You mention there 71 Pm resistance genes discovered in wheat so far. I was wondering what are their locations to get an idea about the Pm resistance genetic architecture. Additionally, many GWAS and QTL mapping studies have been performed w.r.t Pm resistance in wheat. You should have provided some references in that respect.

Line 240: You have mentioned that there are frequent mutations in the pathogen of Pm. Can you please provide some evidence or reasons as to why that is happening. For example, is it because of climate change or sexual reproduction or alternate hosts or because of life cycle of Bgt.

Another important thing is that quite often, disease resistance (for example in your case the new gene of Pm) is achieved at the cost of yield penalty. Have you any idea about the yield penalty that this gene might bring? I mean any field testing you have performed to check the performance of this gene in real time field scenario and comparing the yields with checks.

Last point is that your discussion is 550 words. Its word length need to be increase because it is the crux of the research and it is now a common practice that people are getting more and more interested in going through discussions only.

Best regards, 

Author Response

Dear Reviewer,

Thanks very much for taking your time to review this manuscript. I really appreciate all your comments and suggestions! Please find my itemized responses in below and my revisions/corrections in the re-submitted files.

Sincerely,

Zhenying Wang

Appended to this letter is our point-by-point response to the comments raised by the reviewers. All changes in the text are highlighted in different colors.

(1) Line 71: You mention there 71 Pm resistance genes discovered in wheat so far. I was wondering what are their locations to get an idea about the Pm resistance genetic architecture. Additionally many GWAS and QTL mapping studies have been performed w.r.t Pm resistance in wheat. You should have provided some references in that respect.

Answer: Thank you for the reviewer’s suggestion, we have modified introduction.  The detailed contents were “Research findings have demonstrated that Pm genes are distributed across multiple loci on all wheat chromosomes, with chromosome arms ‌2BS/2BL‌ harboring at least eight Pm genes (Pm6, Pm26, Pm33, Pm42, Pm49, Pm52, Pm63), establishing it as a hotspot for disease resistance breeding. Additionally, contributions from the Aegilops genus (e.g., Aegilops geniculata, Aegilops tauschii) have enriched the diversity of resistance sources in cultivated wheat through the introduction of genes such as Pm12, Pm13, Pm2, and Pm19. Furthermore, ‌Jaegle, B et al. employed a k-mer-based genome-wide association studies (GWAS) approach‌, which not only validated known resistance genes (e.g., Pm2, Pm4) but also ‌discovered 27 novel potential resistance quantitative trait loci (QTL). ‌Ramandeep Kaur et al. utilized GWAS to identified ‌96 candidate genes‌ associated with disease resistance/host-pathogen interactions‌.” The changes were highlighted in purple color.

(2) Line 240: You have mentioned that there are frequent mutations in the pathogen of Pm. Can you please provide some evidence or reasons as to why that is happening. For example, is it because of climate change or sexual reproduction or alternate hosts or because of life cycle ofBgt.

Answer: According to the reviewer’s suggestion, we have revised the description. The detailed contents were “Due to the strong selection for virulent pathogen mutants under agricultural conditions, the emergence and spread of new highly virulent pathogen strains have led to the gradual breakdown of resistance in bred and deployed wheat varieties.” The changes were highlighted in purple color.

(3) Another important thing is that quite often, disease resistance (for example in your case the new gene of Pm) is achieved at the cost of yield penalty. Have you any idea about the yield penalty that this gene might bring? l mean any field testing you have performed to check the performance of this gene in real time field scenario and comparing the yields with checks.

Answer: According to the reviewer’s suggestion, we have added the related descriptions in paragraph 2 and 5 of discussion. The changes were highlighted in green color.

(4) Last point is that your discussion is 550 words. lts word length need to be increase because it is the crux of the research and it is now a common practice that people are getting more and more interested in going through discussions only.

Answer: Thank you for the reviewer’s suggestion, we have fully revised the discussion. The changes were highlighted in purple, green, and orange color.

   The other modifications in PDF file were all revised. The changes were highlighted in purple color.

Round 2

Reviewer 1 Report

Comments and Suggestions for Authors

The manuscript has been significantly improved and all my concerns have been addressed.

Reviewer 4 Report

Comments and Suggestions for Authors

Dear Authors,

Many thanks for refining your MS. I have no further queries.